# The Development of *Populus alba* L. and *Populus tremula* L. Species Specific Molecular Markers Based on 5S rDNA Non-Transcribed Spacer Polymorphism

**Oleg S. Alexandrov \*** and **Gennady I. Karlov**

Laboratory of Plant Cell Engineering, All-Russia Research Institute of Agricultural Biotechnology, Timiryazevskaya 42, Moscow 127550, Russia; karlovg@gmail.com
\* Correspondence: alexandrov@iab.ac.ru; Tel.: +7-499-976-6544

**Abstract:** The *Populus* L. genus includes tree species that are botanically grouped into several sections. This species successfully hybridizes both in the same section and among other sections. Poplar hybridization widely occurs in nature and in variety breeding. Therefore, the development of poplar species' specific molecular markers is very important. The effective markers for trees of the *Aigeiros* Duby section have recently been developed using the polymorphism of 5S rDNA non-transcribed spacers (NTSs). In this article, 5S rDNA NTS-based markers were designed for several species of the *Leuce* Duby section. The *alb9* marker amplifies one fragment with the DNA matrix of *P. alba* and *P. × canescens* (natural hybrid *P. alba × P. tremula*). The *alb2* marker works the same way, except for the case with *Populus bolleana*. In this case, the amplification of three fragments was observed. The *tremu1* marker amplification was detected with the DNA matrix of *P. tremula* and *P. × canescens*. Thus, the developed markers may be applied as a useful tool for *P. alba*, *P. tremula*, *P. × canescens*, and *P. bolleana* identification in various areas of plant science such as botany, dendrology, genetics of populations, variety breeding, etc.

**Keywords:** poplars; molecular markers; non-transcribed spacers; 5S ribosomal DNA; interspecies hybridization; species identification

## 1. Introduction

The *Populus* L. genus includes about three dozen tree species, which are mainly distributed in temperate and subtropical latitudes of the Northern Hemisphere [1]. These species are grouped into the following botanical sections: *Turanga* Bge., *Leucoides* Spach., *Aigeiros* Duby, *Tacamahaca* Spach., and *Populus* (syn. *Leuce* Duby) [2]. Due to their rapid growth, suitable wood, ease of vegetative propagation, and decorative qualities, various types of poplar are widely used for landscaping, protective afforestation, plantation forestry, construction, and other industries [3]. The high economic importance of poplars has increased scientific interest in them. In different countries, several poplar genomes have been recently sequenced [4–8].

The *Populus* species successfully hybridizes both in the same section and among other sections [9]. Poplar hybridization is widespread in nature. Also, it is effectively used in poplar variety breeding [10–14]. However, identifying hybrids and parents by analyzing their morphological characteristics is often difficult. For example, there are environmental influences on their morphology and differences between the juvenile (in hybrid seedlings) and mature (in parent trees) characteristics of poplars [15]. Thus, the development of biochemical and DNA molecular markers has been actively applied to work with poplar species and their hybrids [16,17]. Different types of DNA markers were previously developed, though microsatellite and single nucleotide polymorphism (SNP) markers

are the most commonly used [18–21]. Also, 5S rDNA non-transcribed spacer (NTS) based markers (sequence-characterized amplified region (SCAR) markers) have been recently designed for the identification of some poplars (*P. nigra*, *P. deltoides*, and *P.* × *canadensis*) in the *Aigeiros* section [22].

The 5S rDNA arrays are organized as typical satellite repeats in high plants and animals. In this case, the repeated monomer consists of two parts. The first part is a 120 bp region that codes 5S rRNA. This region is highly conservative. The second part is an NTS, which can be of a different length and nucleotide sequence in different species [23–25]. In poplars, several NTSs were previously sequenced and studied [26–28]. The observed polymorphisms in these NTSs have great potential for species-specific marker development.

In this article, 5S rDNA NTS based markers were used to identify some poplar species in the *Leuce* section. As a useful tool, these markers may be applied in botanical, dendrological, genetic, and breeding research with *P. alba*, *P. tremula*, *P.* × *canescens*, *P. bolleana*, and their hybrids.

## 2. Materials and Methods

### 2.1. Plant Material and DNA Isolation

Twenty-seven poplar trees from different regions of Russia were used as the plant material in this study. The species identification of these samples was conducted by analyzing their morphological characteristics (features of leaves, escapes, fruits, crowns, etc.) according to the keys and descriptions in References [29–33]. Additionally, the *Aigeiros* samples were tested with molecular markers [22]. Young poplar leaves were collected from all samples (Table 1) and used for DNA extraction according to Doyle and Doyle's (1990) protocol [34], with some modifications [35]. After equalization of the concentrations with the help of a NanoDrop® ND-1000 (Thermo Fisher Scientific, Waltham, MA, USA), the quality of the DNA samples was tested by PCR with a universal primer pair based on the 5S rRNA gene (5S1: 5'-GGATGGGTGACCTCCCGGGAAGTCC-3'; 5S2: 5'-CGCTTAACTGCGGAGTTCTGATGGG-3') [36].

### 2.2. Analysis of Sequences and Primer Design

Nine 5S rDNA sequences, including NTSs of *P. alba* and *P. tremula*, were taken from GenBank (*P. alba*: AJ843770, AJ843771, AJ843772, KU994877, KU994878; *P. tremula*: KU994872, KU994873, KU994874, AJ843814). The manipulation, alignment of sequences and identity calculations were carried out in the GenDoc software [37]. The low identity NTSs were used to study their polymorphic regions. Processing of the polymorphism analysis was described in detail by the authors in Reference [22]. The *P. alba* NTS fragments with a high level of polymorphism were selected for the development of the *alb2* and *alb9* markers. The *tremu1* marker was created as follows. The *P. tremula* NTS fragment with a high level of polymorphism was used for the *tremu1* forward primer design. The 120 bp sequence of the 5S rRNA gene was used to design the *5Srev* revers primer. All designed primer pairs were checked with a Multiple Primer Analyzer (Thermo Fisher Scientific, Waltham, MA, USA; see link in Figure S1). The self-dimers and cross-dimer free primers (Table 2) were synthesized by ZAO "Evrogen" (Moscow, Russia) and ZAO "Synthol" (Moscow, Russia).

**Table 1.** Poplar tree samples and their coordinates.

| Species | Parents of Hybrid | Sample Name | Co-ordinates |
|---|---|---|---|
| *P. alba* L. | - | tree#1 | 55°81′46.58″ 37°55′98.41″ |
| | | tree#2 | 44°22′78.85″ 38°89′55.87″ |
| | | tree#3 | 44°64′65.02″ 39°14′36.91″ |
| | | tree#4 | 45°29′43.82″ 36°42′96.96″ |
| | | tree#5 | 50°44′89.88″ 39°63′36.73″ |
| | | tree#6 | 55°83′62.65″ 37°56′70.35″ |
| *P. tremula* L. | - | tree#7 | 55°83′50.41″ 37°55′53.78″ |
| | | tree#8 | 56°23′92.56″ 38°11′17.26″ |
| | | tree#9 | 56°06′58.33″ 37°90′21.71″ |
| | | tree#10 | 55°90′05.39″ 37°56′67.05″ |
| | | tree#11 | 55°41′81.88″ 37°84′72.91″ |
| | | tree#12 | 67°08′07.96″ 32°86′88.48″ |
| *P.* × *canescens* (Aiton) Sm. | *P. alba* L. × *P. tremula* L. | tree#13 | 55°45′11.99″ 36°94′16.11″ |
| | | tree#14 | 55°72′38.02″ 37°56′49.92″ |
| | | tree#15 | 55°85′23.54″ 37°54′33.94″ |
| *P. bolleana* Lauche | | tree#16 | 45°29′56.17″ 36°43′13.14″ |
| | | tree#17 | 45°30′05.94″ 36°42′79.16″ |
| | | tree#18 | 45°30′04.41″ 36°42′81.74″ |
| *P. nigra* L. | | tree#19 | 50°83′42.96″ 39°39′21.76″ |
| *P. deltoides* Bartr. ex Marshall | - | tree#20 | 55°83′51.72″ 37°55′55.98″ |
| *P.* × *canadensis* Moench. | *P. deltoides* Bartr. ex Marshall × *P. nigra* L. | tree#21 | 55°82′83.67″ 37°57′47.36″ |
| *P. trichocarpa* Torr. et A. Gray | - | tree#22 | 55°83′48.51″ 37°55′55.97″ |
| *P. maximowiczii* Henry | - | tree#23 | 55°83′52.52″ 37°55′53.57″ |
| *P. simonii* Can. | - | tree#24 | 55°83′52.74″ 37°55′54.93″ |
| *P. candicans* Ait. | - | tree#25 | 55°83′53.31″ 37°55′52.29″ |
| *P.* × *moskoviensis* R. I. Schrod. | *P. suaveolens* Fish. (syn. *P. maximowiczii* Henry) × *P. laurifolia* Ldb. | tree#26 | 55°83′53.36″ 37°55′54.88″ |
| *P.* × *berolinensis* K. Koch. | *P. laurifolia* Ldb. × *P. nigra* L. | tree#27 | 55°83′51.80″ 37°55′55.15″ |

**Table 2.** The primer sequences and expected lengths of the PCR products.

| Primer Name | Sequence | PCR Product Length, bp |
|---|---|---|
| *alb2-f* | 5′-TTTTGCCGTTTTCC-3′ | 110 |
| *alb2-r* | 5′-AATCGCCCGGGAAAGGAAA-3′ | |
| *alb9-f* | 5′-TCGGAGTAGCGATTCACAGC-3′ | 126 |
| *alb9-r* | 5′-GTTTGCGTCGGACCATAACA-3′ | |
| *tremu1* | 5′-AGCCTCCCGCTGGG-3′ | 113 |
| *5Srev* | 5′-CGCTTAACTGCGGAGT-3′ | |

*2.3. PCR and Electrophoresis*

The PCR conditions for the *alb2* and *alb9* markers were as follows: 94 °C for 5 min; 35 cycles of 94 °C for 20 s, 60 °C for 20 s, and 72 °C for 20 s; then, 72 °C for 10 min. For the *tremu1* marker, the number of cycles was 30, and the annealing temperature was 68 °C. The other parameters were the same. The PCR products were separated by electrophoresis on 2.5% agarose gel at 6 V/cm in a 0.5 M TBE buffer using a Sub-Cell Model 192 camera (Bio-Rad, Hercules, CA, USA). In electrophoresis, the marker of molecular length "100 bp DNA Ladder" (Jena Bioscience GmbH, Jena, Germany) was used. The results of the electrophoresis were visualized and photographed using the GelDoc XR Plus (Bio-Rad, Hercules, CA, USA) gel documentation system.

## 3. Results

The *P. alba* and *P. tremula* NTSs were isolated from the collected sequences (*P. alba*: AJ843770, AJ843771, AJ843772, KU994877, KU994878 and *P. tremula*: KU994872, KU994873, KU994874, AJ843814 (see Figure S2)). All NTSs had different lengths. For *P. alba*, the length ranged from 153 to 323 bp. In *P. tremula*, the length ranged from 132 to 477 bp. The NTSs were aligned (Figure S3) and compared based on their levels of homology (Table S4). It was found that the *P. alba* NTSs had a 39–93% level of homology among themselves. For the *P. tremula* NTSs, the value of the indicator was 15–76%. The NTS homology between the two studied species was 18–48%. Some NTSs with an 18% level of homology in any case (in *P. alba*, AJ843770 and AJ843772; in *P. tremula*, KU994872) were selected for a detailed analysis of their polymorphisms.

In the analysis of the polymorphism between the *P. alba* AJ843770, AJ843772 NTSs, and all *P. tremula* NTSs, two alignments were conducted. The first alignment (with *P. alba* AJ843770 NTS) consisted of 336 columns (Table S5). The analysis of the polymorphism in the ten-column fragments did not show any regions with high levels of polymorphism (>90%). However, the region between the 288 and 318 points of the alignment (indicated with orange letters in Table S5) was evaluated to be promising for primer design because it had four 1–5 bp deletions (indicated with yellow colored cells) in the AJ843770 NTS. As a result, this region was changed for the PCR test (*alb2-r* primer, Table 2). The forward primer for *alb2-r* (*alb2-f* primer, Table 2) was designed at the beginning of the AJ843770 NTS (indicated with red letters in Table S5). The second alignment (with *P. alba* AJ843772) consisted of 471 columns (Table S6). In this case, the analysis of polymorphism in the ten-column fragments detected four extended regions (between 1 and 40, 105 and 115, 127 and 136, and 316 and 339 points of the alignment) with a high level of polymorphism (>90%). The fourth region (indicated with orange letters in Table S6) was changed as the basis for primer design (*alb9-r* primer, Table 2). The forward primer for *alb9-r* (*alb2-f* primer, Table 2) was designed in the region located in front of the 316–339 region (indicated with red letters in Table S6).

In the analysis of the polymorphism between the *P. tremula* KU994872 NTS and all *P. alba* NTSs, the 509-column alignment (Table S7) showed two extended regions (between 234 and 334, 423 and 486 points of the alignment) with a 56–76% level of polymorphism (regions with a high level of polymorphism (>90%) were not detected). One of the second extended regions (indicated with red letters in Table S7) was selected as the basis for the primer design (*tremu1* primer, Table 2). The reverse

primer for *tremu1* (*5Srev* primer, Table 2) was designed in the region located in the middle of the 5S rRNA gene sequence.

Thus, three pairs of the designed primers were synthesized for the PCR test. The test was conducted in two stages. In the first stage, only the samples of the *Leuce* poplars were used (Figure 1a–c). In the second stage, the markers were checked with the DNA matrices of the poplars from other sections (Figure S8a–c). The PCR with the *alb2* marker (Figure 1a) resulted in the amplification of one 110 bp fragment in the case with all *P. alba* and *P. × canescens* samples. This fragment was also amplified in the *P. bolleana* profile. The profile also included two added fragments (approximately 85 and 135 bp). As for all *P. tremula* samples, there was no amplification. The results of the *alb9* marker work (Figure 1b) are as follows. The amplification of the expected 126 bp fragment was detected in cases with all the *P. alba*, *P. bolleana*, and *P. × canescens* samples. In all *P. tremula* samples, the amplification was absent, as was the case with the *alb2* marker. The work of the *tremu1* marker consisted of amplifying the 113 bp fragment in cases with all the *P. tremula* and *P. × canescens* samples. In cases with all *P. alba* and *P. bolleana* samples, no fragment was amplified (Figure 1c). Also, the second stage of the PCR test showed the absence of the amplification with all studied markers in the cases with poplars of other sections (Figure S8a–c).

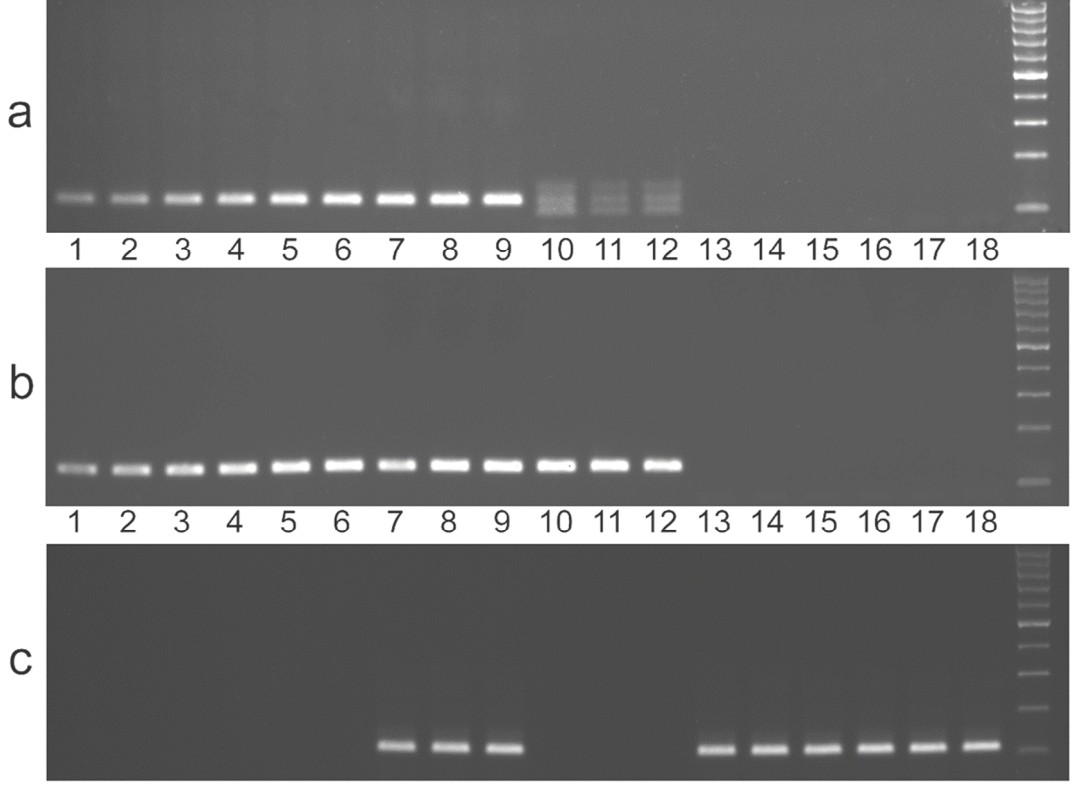

**Figure 1.** The results of the first stage of the PCR test: (**a**) With the *alb2* marker; (**b**) with the *alb9* marker; and (**c**) with the *tremu1* marker. The numbers in all panels correspond to the following samples: 1—*P. alba* L. tree#1; 2—*P. alba* L. tree#2; 3—*P. alba* L. tree#2; 4—*P. alba* L. tree#4; 5—*P. alba* L. tree#5; 6—*P. alba* L. tree#6; 7—*P. × canescens* (Aiton) Sm. tree#13; 8—*P. × canescens* (Aiton) Sm. tree#14; 9—*P. × canescens* (Aiton) Sm. tree#15; 10—*P. bolleana* Lauche tree#16; 11—*P. bolleana* Lauche tree#17; 12—*P. bolleana* Lauche tree#18; 13—*P. tremula* L. tree#7; 14—*P. tremula* L. tree#8; 15—*P. tremula* L. tree#9; 16—*P. tremula* L. tree#10; 17—*P. tremula* L. tree#11; 18—*P. tremula* L. tree#12.

## 4. Discussion

In this article, all sequences of *P. alba* and *P. tremula* NTSs included in GenBank were analyzed, as in the previous study with *P. nigra* and *P. deltoides* [22]. NTSs of these black poplar species were

previously observed to separate into two classes [22,26]. For *P. alba* and *P. tremula* NTSs, such a clear separation was not observed. All studied NTSs had different lengths. The level of the homology among the NTSs of one species was also different. In general, the studied *P. tremula* NTS sequences were more homologous to each other than the *P. alba* NTSs. The KU994872 and KU994873 NTSs were the most alike (with a 93% level of homology). They were distinguished by seven single nucleotide substitutions and one 14 bp deletion only (these mutations are indicated with a red shade in Figure S9). Despite this example, the intraspecies level of NTS homology in *P. tremula* and *P. alba* was generally lower than that in *P. nigra* and *P. deltoides* (as previously reported [22], this indicator was 71–98% among these species of black poplars). In the two compared cases, the interspecies level of NTS homology was also lower, with the exception of several NTS pairs, which had a similar value for this indicator. Thus, the potential for successful species–specific marker development was originally rated as quite high.

However, it is worth noting that, in practice, the development of the *alb2* and *tremu1* markers experienced some difficulties. In the *alb2* development, the AJ843770 NTS had a small length (in comparison with the other analyzed NTSs), and its alignment included many gaps in the bottom line (the line with the sequence, which is rated). Since the gap in this line corresponds to a zero value for the polymorphism level at this point of the alignment, the polymorphism level of the ten-column fragments with this point decreased. As a result, ten-column polymorphism was not high when aligned with the AJ843770 NTS (the level of polymorphism was considered high if it was >90%, as in [22]). This problem was not observed in the marker development of black poplars, in which the length differences between the rated NTS and the others were not large [22]. In sum, the problem with the AJ843770 NTS was solved by designing a primer based on the short region, with a large number of deletions. This approach may be a useful solution when the markers are developed in similar conditions. In the *tremu1* development, there were two problems. The first problem consisted of the presence of two short (in comparison with the others) NTS sequences (AJ843770 and KU994878) in the alignment, which were compared to the rated sequence (KU994872). Since the gap in such lines (all lines except the bottom line) was considered as a polymorphic nucleotide, the polymorphism level of the ten-column fragments with this point increased. When other nucleotides in this column are polymorphic, there is no problem. However, if other nucleotides in this column coincide (or at least two of them—the first in the bottom line and the second in any other line), then the polymorphism level of the ten-column fragments with this point become overpriced, and the fragments may not be suitable for marker development. This situation was observed for most parts of the 234–334 and 423–486 extended regions in the *tremu1* alignment. As a solution to this problem, short sequences may be excluded from the alignment. In the test with these *tremu1* extended regions, the exclusion of AJ843770 and KU994878 NTSs forming the alignment decreased the polymorphism level of the ten-column fragments (Table S10). The second problem in the *tremu1* development is the following. The selected region for primer production stood out by the level of polymorphism in that alignment place. However, there were regions in other places of alignment, which coincided with the 5′-part of the selected region (these coincidences are indicated with a red shade in Figure S11). Such regions are occasionally masked in the process of alignment construction and may be detected with the help of the Basic Local Alignment Search Tool (BLAST). As a solution to this problem in the *tremu1* development, the following approach was used. Since coincidences were detected in the 5′-part only, the primer was designed with a forward orientation, and the annealing temperature was increased as much as possible.

In the cases with the *P. alba*, *P. tremula*, and *P.* × *canescens* samples, all markers were amplified as expected. The *alb2* and *alb9* amplification with no *tremu1* amplification was observed in the *P. alba* samples. The *tremu1* amplification with no *alb2* and *alb9* amplification was found in the *P. tremula* samples. The amplification of *alb2*, *alb9*, and *tremu1* was detected in *P.* × *canescens* (natural hybrid *P. alba* × *P. tremula*) samples. Thus, an effective molecular marker system for the rapid identification of these two species and their hybrids was created. This result may have importance in the genetics of populations, as these plants are often used as model objects. Usually, SSR marker sets are applied in

such experiments [19,20]. The use of the presented marker system may decrease the volume of PCRs, thereby increasing the speed of research and reducing their cost.

Work of the presented markers in other plants of the *Leuce* section has a significant interest for botanists. In this article, the *P. bolleana* (the white poplar with a pyramidal crown, which is considered by some scientists as a form or synonym of *P. alba* [38,39]) samples were tested with the developed markers. The results of the *alb9* and *tremu1* amplification with the *P. bolleana* samples were exactly the same as those for the *P. alba* samples. However, for the *alb2* marker, an amplification of two added fragments was detected in the profile. The reason for this result may be revealed via a thorough study of *P. bolleana* 5S rDNA NTSs. Nevertheless, the *alb2* marker application will be useful, for example, in checking of *P. alba* and *P. bolleana* herbarium materials.

The PCR experiments with poplar trees of other sections showed no amplification for each of the presented markers. This fact shows that these markers are *Leuce* section specific. Thus, the developed markers may be a useful tool for *P. alba*, *P. tremula*, *P.* × *canescens*, and *P. bolleana* identification in various areas of plant science, such as botany, dendrology, the genetics of populations, and variety breeding.

**Supplementary Materials:** The following are available online at http://www.mdpi.com/1999-4907/10/12/1092/s1. Figure S1: The Multiple Primer Analyzer link; Figure S2: The isolated sequences of all *P. alba* and *P. tremula* 5S rDNA NTSs; Figure S3: The alignment of all described *P. alba* and *P. tremula* 5S rDNA NTSs; Table S4: The levels of homology among the described *P. alba* and *P. tremula* 5S rDNA NTSs; Table S5: The results of the polymorphism level analysis in ten-column fragments of the AJ843770 *P. alba* 5S rDNA NTS; Table S6: The results of the polymorphism level analysis in ten-column fragments of the AJ843772 *P. alba* 5S rDNA NTS; Table S7: The results of the polymorphism level analysis in ten-column fragments of the KU994872 *P. tremula* 5S rDNA NTS; Figure S8: The results of the second step of the PCR test; Figure S9: The alignment of the KU994872 and KU994873 *P. tremula* 5S rDNA NTSs; Table S10: The results of the polymorphism level analysis in ten-column fragments of the KU994872 *P. tremula* 5S rDNA NTS without AJ843770 and KU994878 NTSs; Figure S11: The results of the coincidence search of the selected region in the KU994872 *P. tremula* 5S rDNA NTS alignment.

**Author Contributions:** O.S.A. and G.I.K. conceived and designed the experiments and formulated the discussion; O.S.A. performed the experiments, analyzed the data, and wrote the paper.

**Funding:** This work was carried out as part of the state assignment of the Ministry of Science and Higher Education of the Russian Federation No. 0574-2019-0002 (AAAA-A18-118051890089-0).

**Acknowledgments:** The authors are grateful to A.N. Sorokin (Department of Tropical and Subtropical Plants, Main Botanical Garden named by N.V. Tsitsin, RAS), T.G. Machrova (Department of Breeding, Genetics and Dendrology, Mytischi Branch of Bauman Moscow State Technical University), T.A. Feodorova (Department of Higher Plants, Biology Faculty, Lomonosov Moscow State University), and N.C. Potapenko (Research Institute Botanical Garden, Nizhni Novgorod State University named after N.I. Lobachevsky) for their help in plant material collection.

**Conflicts of Interest:** The authors declare no conflict of interest.

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
