# Peer review of "The Development of Populus alba L. and Populus tremula L. Species Specific Molecular Markers Based on 5S rDNA Non-Transcribed Spacer Polymorphism"

_forests, doi:10.3390/f10121092_

Round 1

Reviewer 1 Report

This manuscript is a worthwhile description of the development of species specific molecular markers for Populus alba and Populus tremula. The topic fits well within the scope of the journal, and presentation of the information is appropriate for a short communication (as the authors have submitted). I do not have any substantive concerns, but there is a moderate amount of English editing that needs to be done before publication. I suggest the authors work with an English editor to polish the paper. 

From a technical standpoint, I have a few suggestions.

The authority for section Aigeiros is Duby NOT Daby. In multiple places, the authors use the phrase "between some sections" - this should state "among some sections" when talking about more than two groups. Eliminate keywords that are redundant with the title. Line 28: list the actual number of poplar species - see Isebrands and Richardson (2014) Poplars and Willows: Trees for Society and the Environment Line 34: enlargement is an awkward word choice Line 39: what are the different reasons? I only see one listed. Table 1: P. x canadensis is a P. deltoides x P. nigra hybrid, not P. nigra x P. deltoides (remember, the female parent is always listed first). Table 1: P. suaveolens is currently listed as a subspecies of P. maximowiczii - so I would just use the latter instead of P. suaveolens in the table. Line 120: spelling - reverse

Author Response

Dear colleague!

Thank you very much for your review. According to your recommendation, we polished the text with an English editor from mdpi service.

Also, your suggestions have allowed us to improve the article. In particular:

Suggestion 1. The authority for section Aigeiros is Duby NOT Daby.

Answer: It is our mistake. We fixed it in all the necessary places.

Suggestion 2. In multiple places, the authors use the phrase "between some sections" - this should state "among some sections" when talking about more than two groups.

Answer: We checked all the indicated places and fixed the mistakes.

Suggestion 3. Eliminate keywords that are redundant with the title.

Answer: We have been corrected the “keywords” part.

Suggestion 4. Line 28: list the actual number of poplar species - see Isebrands and Richardson (2014) Poplars and Willows: Trees for Society and the Environment

Answer: We changed this sentence and inserted a link to the book you recommended.

Suggestion 5. Line 34: enlargement is an awkward word choice Line 39: what are the different reasons? I only see one listed. Table 1: P. x canadensis is a P. deltoides x P. nigra hybrid, not P. nigra x P. deltoides (remember, the female parent is always listed first).

Answer: We fixed this mistake.

Suggestion 6. Table 1: P. suaveolens is currently listed as a subspecies of P. maximowiczii - so I would just use the latter instead of P. suaveolens in the table.

Answer: Unfortunately, we cannot accept this suggestion in full. As we consider, the issue about P. suaveolens and P. maximowiczii is not finally resolved. In other our experiments, we observed the differences between these species in 5S rDNA. Additionally, P. × moskoviensis had been described by author R. I. Schroder as hybrid P. suaveolens Fish. and P. laurifolia Ldb. And we do not found literature sources when it would be described as hybrid P. maximowiczii and P. laurifolia Ldb. As a compromise, we resolved to write “P. suaveolens Fish. (syn. P. maximowiczii Henry) × P. laurifolia Ldb”.

Suggestion 7. Line 120: spelling – reverse.

Answer: We fixed this mistake.

Reviewer 2 Report

The problem of identifying Populus species is an urgent and difficult task since the presence of many intra- and intersectional hybrids requires high qualification and considerable experience in determining the species according to morphological features.

In the study by Alexandrov and Karlov, molecular markers have been proposed for the identification of P. alba, P. tremula, P. × canescens, and P. bolleana. The work was performed at a good methodological level, the approaches used are described in detail in the manuscript that allows other scientists to use them in phylogenetic studies of the Populus genus, as well as to identify species in genetic collections and nurseries.

The disadvantage of the manuscript is an insufficiently detailed description of the plant material used, in particular, how the morphological identification of species was performed. The work represents a completed study and, after introducing additional information on the plant material, it can be accepted for publication in the Forests journal.

Author Response

Dear colleague!

Thank you very much for your good review. The recommended description of our plant material was added in the article (line 58-62). Also, we indicate the literature sources (references 29-33) that we used for species identification.
